# Scalable Normalizing Flows Enable Boltzmann Generators for Macromolecules

## Abstract

The Boltzmann distribution of a protein provides a roadmap to all of its functional states. Normalizing flows are a promising tool for modeling this distribution, but current methods are intractable for typical pharmacological targets; they become computationally intractable due to the size of the system, heterogeneity of intramolecular potential energy, and long-range interactions. To remedy these issues, we present a novel flow architecture that utilizes split channels and gated attention to efficiently learn the conformational distribution of proteins defined by internal coordinates. We show that by utilizing a 2-Wasserstein loss, one can smooth the transition from maximum likelihood training to energy-based training, enabling the training of Boltzmann Generators for macromolecules. We evaluate our model and training strategy on villin headpiece HP35(nle-nle), a 35-residue subdomain, and protein G, a 56-residue protein. We demonstrate that standard architectures and training strategies, such as maximum likelihood alone, fail while our novel architecture and multi-stage training strategy are able to model the conformational distributions of protein G and HP35.

## 1 Introduction

The structural ensemble of a protein determines its functions. The probabilities of the ground and metastable states of a protein at equilibrium for a given temperature determine the interactions of the protein with other proteins, effectors, and drugs, which are keys for pharmaceutical development. However, enumeration of the equilibrium conformations and their probabilities is infeasible. Since complete knowledge is inaccessible, we must adopt a sampling approach. Conventional approaches toward sampling the equilibrium ensemble rely on Markov-chain Monte Carlo or molecular dynamics (MD). These approaches explore the local energy landscape adjacent a starting point; however, they are limited by their inability to penetrate high energy barriers. In addition, MD simulations are expensive and scale poorly with system size. This results in incomplete exploration of the equilibrium conformational ensemble.

In their pioneering work, Noé et al. (2019) proposed a normalizing flow model (Rezende & Mohamed, 2015), that is trained on the energy function of a many-body system, termed Boltzmann generators (BGs). The model learns an invertible transformation from a system's configurations to a latent space representation, in which the low-energy configurations of different states can be easily sampled. As the model is invertible, every latent space sample can be back-transformed to a system configuration with high Boltzmann probability, i.e., $p(\mathbf{x}) \propto e^{-u(\mathbf{x})/(kT)}$.

A normalizing flow-based generative model is constructed by a sequence of invertible transformations (Rezende & Mohamed, 2015). BGs typically employ flow models because they can be sampled from efficiently and they describe a tractable probability density function. This allows us to employ reverse KL divergence training since we can compute an unnormalized density for the target Boltzmann distribution (Noé et al., 2019; Wirnsberger et al., 2022; Köhler et al., 2021).

BGs in the literature have often struggled with even moderate-sized proteins, due to the complexity of conformation dynamics and scarcity of available data. Most works have focused on small systems like alanine dipeptide (22 atoms) (Köhler et al., 2021; Midgley et al., 2022; Invernizzi et al., 2022). To date, only two small proteins, BPTI and bromodomain, have been modeled by BGs. Noé et al. (2019) trained a BG for BPTI, a 58 amino acid structure, at all-atom resolution. Unfortunately, the

training dataset used is licensed by DESRES (Shaw et al., 2010) and not open-source. No works since have shown success on proteins of similar size at all-atom resolution or reported results for BPTI. Mahmoud et al. (2022) trained a BG for bromodomain, a 100 residue protein, with a SIRAH coarse-grained representation. However, drug design applications require much finer resolution than resolvable by SIRAH. A thorough review of related works is detailed in Appendix A.

The limited scope of flow model BG applications is due to the high computational expense of their training process. Their invertibility requirement limits expressivity when modeling targets whose supports have complicated topologies (Cornish et al., 2019), necessitating the use of many transformation layers. Another hurdle in scaling BGs is that proteins often involve long-range interactions; atoms far apart in sequence can interact with each other. In this work, we present a new BG method for general proteins with the following contributions:

- We use a global internal coordinate representation with fixed bond-lengths and side-chain angles. From a global structure and energetics point-of-view, little information is lost by allowing side-chain bonds to only rotate. Such a representation not only reduces the number of variables but also samples conformations more efficiently than Cartesian coordinates (Noé et al., 2019; Mahmoud et al., 2022).

- The global internal coordinate representation is initially split into a backbone channel and a side-chain channel. This allows the model to efficiently capture the distribution of backbone internal coordinates, which most controls the overall global conformation.

- A new NN architecture for learning the transformation parameters of the coupling layers of the flow model which makes use of gated attention units (GAUs) (Hua et al., 2022) and a combination of rotary positional embeddings (Su et al., 2021) with global, absolute positional embeddings for learning long range interactions.

- To handle global conformational changes, a new loss-function, similar in spirit to the *Fréchet Inception Distance (FID)* (Heusel et al., 2017), is introduced to constrain the global backbone structures to the space of native conformational ensemble.

We show in this work that our new method can efficiently generate Boltzmann distributions and important experimental structures in two different protein systems. We demonstrate that the traditional maximum likelihood training for training flow models is insufficient for proteins, but our multi-stage training strategy can generate samples with high Boltzmann probability.

## 2 BACKGROUND

### 2.1 NORMALIZING FLOWS

Normalizing flow models learn an invertible map $f : \mathbb{R}^d \mapsto \mathbb{R}^d$ to transform a random variable $\boldsymbol{z} \sim q_Z$ to the random variable $\boldsymbol{x} = f(\boldsymbol{z})$ with distribution

$$q_X(\boldsymbol{x}) = q_Z(\boldsymbol{z})|\det(J_f(\boldsymbol{z}))|^{-1}, \tag{1}$$

where $J_f(\boldsymbol{z}) = \partial f/\partial \boldsymbol{z}$ is the Jacobian of $f$. We can parameterize $f$ to approximate a target distribution $p(\boldsymbol{x})$. To simplify notation, we refer to the flow distribution as $q_\theta$, where $\theta$ are the parameters of the flow. If samples from the target distribution are available, the flow can be trained via maximum likelihood. If the unnormalized target density $p(\boldsymbol{x})$ is known, the flow can be trained by minimizing the KL divergence between $q_\theta$ and $p$, i.e., $\mathrm{KL}(q_\theta||p) = \int_X q_\theta(\boldsymbol{x}) \log(q_\theta(\boldsymbol{x})/p(\boldsymbol{x}))d\boldsymbol{x}$.

### 2.2 DISTANCE MATRIX

A protein distance matrix is a square matrix of Euclidean distances from each atom to all other atoms. Practitioners typically use $C\alpha$ atoms or backbone atoms only. Protein distance matrices have many applications including structural alignment, protein classification, and finding homologous proteins (Holm & Sander, 1993; Holm, 2020; Zhu et al., 2023). They have also been used as representations for protein structure prediction algorithms, including the first iteration of AlphaFold (Senior et al., 2019; Xu & Wang, 2019; Hou et al., 2019).

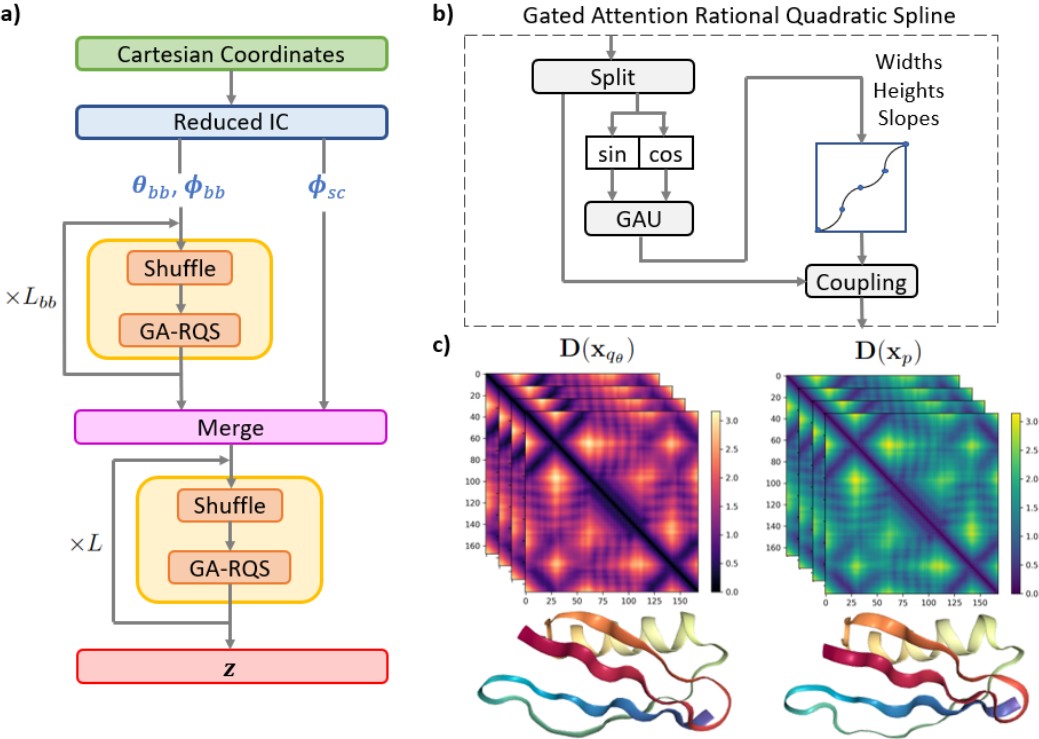

Figure 1: (a) Our split flow architecture. (b) Each transformation block consists of a gated attention rational quadratic spline (RQS) coupling layer. (c) Example structures of protein G from the flow $q_\theta$ (left) and from molecular dynamics simulation $p$ (right). We also show sample distance matrices $\mathbf{D}(\mathbf{x}_{q_\theta})$ and $\mathbf{D}(\mathbf{x}_p)$.

## 2.3 2-WASSERSTEIN DISTANCE

The 2-Wasserstein Distance is a measure of the distance between two probability distributions. Let $P = \mathcal{N}(\boldsymbol{\mu}_P, \boldsymbol{\Sigma}_P)$ and $Q = \mathcal{N}(\boldsymbol{\mu}_Q, \boldsymbol{\Sigma}_Q)$ be two normal distributions in $\mathbb{R}^d$. Then, with respect to the Euclidean norm on $\mathbb{R}^d$, the squared 2-Wasserstein distance between $P$ and $Q$ is defined as

$$W_2(P,Q)^2 = \|\boldsymbol{\mu}_P - \boldsymbol{\mu}_Q\|_2^2 + \text{trace}(\boldsymbol{\Sigma}_P + \boldsymbol{\Sigma}_Q - 2(\boldsymbol{\Sigma}_P\boldsymbol{\Sigma}_Q)^{1/2}). \tag{2}$$

In computer vision, the Fréchet Inception Distance (FID) (Heusel et al., 2017) computes the 2-Wasserstein distance and is often used as an evaluation metric to measure generated image quality.

## 3 SCALABLE BOLTZMANN GENERATORS

### 3.1 PROBLEM SETUP

BGs are generative models that are trained to sample from the Boltzmann distribution for physical systems, i.e., $p(\mathbf{x}) \propto e^{-u(\mathbf{x})/(kT)}$, where $u(\mathbf{x})$ is the potential energy of the conformation $\mathbf{x}$, $k$ is the Boltzmann constant, and $T$ is the temperature. A protein conformation is defined as the arrangement in space of its constituent atoms (Fig. 2), specifically, by the set of 3D Cartesian coordinates of its atoms. Enumeration of metastable conformations for a protein at equilibrium is quite challenging with standard sampling techniques. We tackle this problem with generative modeling. Throughout this work, we refer to $p$ as the ground truth conformation distribution and $q_\theta$ as the distribution parameterized by the normalizing flow model $f_\theta$.

### 3.2 REDUCED INTERNAL COORDINATES

Energetically-favored conformational changes take place via rotations around single chemical bonds while bond vibrations and angle bending at physiologic temperature result in relatively small spatial perturbations (Vaidehi & Jain, 2015). Our focus on near ground and meta-stable states therefore motivates the use of internal coordinates: $N - 1$ bond lengths $d$, $N - 2$ bond angles $\theta$, and $N - 3$ torsion angles $\phi$, where $N$ is the number of atoms of the system (see Fig. 2). In addition, internal coordinate representation is translation and rotation invariant.

A protein can be described as a branching structure with a set of backbone atoms and non-backbone atoms (we will colloquially refer to these as side-chain atoms). Previous works have noted the difficulty in working with internal coordinate representations for the backbone atoms (Noé et al., 2019; Köhler et al., 2022; Mahmoud et al., 2022). This is due to the fact that protein conformations are sensitive to small changes in the backbone torsion angles. Noé et al. (2019) introduced a coordinate transformation whereby the side-chain atom coordinates are mapped to internal coordinates while the backbone atom coordinates are linearly transformed via principal component analysis (PCA) and the six coordinates with the lowest variance are eliminated. However, as mentioned by Midgley et al. (2022), the mapping of vectors onto a fixed set of principal components is generally not invariant to translations and rotations. In addition, PCA suffers from distribution shift.

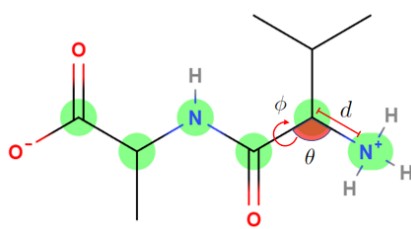

Figure 2: A two residue chain. Hydrogens on carbon atoms are omitted. Backbone atoms are highlighted green. Shown is an example of a bond length $d$, a bond angle $\theta$, and a dihedral/torsion angle $\phi$.

A full internal coordinate system requires $3N - 6$ dimensions where $N$ is the number of atoms. Bond lengths hardly vary in equilibrium distributions while torsion angles can vary immensely. We treat non-backbone bond angles as constant, again replaced by their mean. Heterocycles in the sidechains of Trp, Phe, Tyr and His residues are treated as rigid bodies. Our final representation is

$$\mathbf{x} = [\boldsymbol{\theta}_{bb}, \boldsymbol{\phi}_{bb}, \boldsymbol{\phi}_{sc}],$$

where the subscripts $bb$ and $sc$ indicate backbone and sidechain, respectively. This dramatically reduces the input dimension and keeps the most important features for learning global conformation changes in the equilibrium distribution.

Recent works have adopted similar approaches to reduce the input number of dimensions. Wu et al. (2022) utilize only the backbone torsion and bond angles to represent proteins for novel protein design and generation, while Wang et al. (2022) simply use the backbone torsion angles to represent the polypeptide AiB9 in a similar task of modeling the polypeptide conformational landscape.

### 3.3 TRAINING AND EVALUATION

We train BGs with MD simulation data at equilibrium, i.e., the distribution of conformations is constant and not changing as with, for example, folding. We seed the simulation with energetically stable native conformations. BG training aims to learn to sample from the Boltzmann distribution of protein conformations. We compute the energy of samples generated by our model under the AMBER14 forcefield (Case et al., 2014) and report their mean. In addition, in order to evaluate how well the flow model generates the proper backbone distribution, we define a new measure:

**Definition 3.1** (Distance Distortion). Let $\mathcal{A}_{bb}$ denote the indices of backbone atoms. Define $\mathbf{D}(\mathbf{x})$ as the pairwise distance matrix for the backbone atoms of $\mathbf{x}$. Define $\mathcal{P} = \{(i,j) | i, j \in \mathcal{A}_{bb}$ and $i < j\}$. The distance distortion is defined as

$$\Delta D := \mathbb{E}_{\substack{\mathbf{x}_{q_\theta} \sim q_\theta \\ \mathbf{x}_p \sim p}} \left[ \frac{1}{|\mathcal{P}|} \sum_{(i,j) \in \mathcal{P}} |\mathbf{D}(\mathbf{x}_{q_\theta})_{ij} - \mathbf{D}(\mathbf{x}_p)_{ij}| \right], \tag{3}$$

### 3.4 SPLIT FLOW ARCHITECTURE

We use Neural Spline Flows (NSF) with rational quadratic splines (Durkan et al., 2019) having 8 bins each. The conditioning is done via coupling. Torsion angles $\phi$ can freely rotate and are therefore treated as periodic coordinates (Rezende et al., 2020).

The full architectural details are highlighted in Fig. 1. We first split the input into backbone and sidechain channels:

$$\mathbf{x}_{bb} = [\boldsymbol{\theta}_{bb}, \boldsymbol{\phi}_{bb}], \qquad \mathbf{x}_{sc} = [\boldsymbol{\phi}_{sc}].$$

We then pass the backbone inputs through $L_{bb} = 48$ gated attention rational quadratic spline (GA-RQS) coupling blocks. As all the features are angles in $[-\pi, \pi]$, we augment the features with their mapping on the unit circle. In order to utilize an efficient attention mechanism, we employ gated attention units (GAUs) (Hua et al., 2022). In addition, we implement relative positional embeddings (Shaw et al., 2018) on a global level so as to allow each coupling block to utilize the correct embeddings. The backbone latent embeddings are then concatenated with the side chain features and passed through $L = 10$ more GA-RQS coupling blocks.

### 3.5 MULTI-STAGE TRAINING STRATEGY

Normalizing flows are most often trained by maximum likelihood, i.e., minimizing the negative log likelihood (NLL)

$$\mathcal{L}_{\text{NLL}}(\theta) := -\mathbb{E}_{\mathbf{x} \sim p}[\log q_\theta(\mathbf{x})], \tag{4}$$

or by minimizing the reverse KL divergence [1]:

$$\mathcal{L}_{\text{KL}}(\theta) := \mathbb{E}_{\mathbf{x} \sim q_\theta}[\log(q_\theta(\mathbf{x})/p(\mathbf{x}))]. \tag{5}$$

In the BG literature, minimizing the KL divergence is often referred to as "training-by-energy", as the expression can be rewritten in terms of the energy of the system. The reverse KL divergence suffers from mode-seeking behavior, which is problematic when learning multimodal target distributions. While minimizing the NLL is mass-covering, samples generated from flows trained in this manner suffer from high variance. In addition, for larger systems, maximum likelihood training often results in high-energy generated samples.

In their seminal paper, Noé et al. (2019) used a convex combination of the two loss terms, in the context of BGs, in order to both avoid mode-collapse and generate low-energy samples. However, for larger molecules, target evaluation is computationally expensive and dramatically slows iterative training with the reverse KL divergence objective. In addition, during the early stages of training, the KL divergence explodes and leads to unstable training. One way to circumvent these issues is to train with the NLL loss, followed by a combination of both loss terms. Unfortunately, for larger systems, the KL term tends to dominate and training often get stuck at non-optimal local minima. In order to remedy these issues, we consider a sequential training scheme, whereby we smooth the transition from maximum likelihood training to the combination of maximum likelihood and reverse KL divergence minimization.

(1) As mentioned previously, we first train with maximum likelihood to convergence.

(2) Afterward, we train with a combination of the NLL and the 2-Wasserstein loss with respect to distance matrices of the backbone atoms:

$$\mathcal{L}_{\text{W}}(\theta) := \|\boldsymbol{\mu}_{q_\theta} - \boldsymbol{\mu}_p\|_2^2 + \text{trace}(\boldsymbol{\Sigma}_{q_\theta} + \boldsymbol{\Sigma}_p - 2(\boldsymbol{\Sigma}_{q_\theta}\boldsymbol{\Sigma}_p)^{1/2}), \tag{6}$$

where

$$\boldsymbol{\mu}_p := \mathbb{E}_{\mathbf{x} \sim p}[\mathbf{x}_{bb}], \quad \boldsymbol{\Sigma}_p := \mathbb{E}_{\mathbf{x} \sim p}[(\mathbf{x}_{bb} - \boldsymbol{\mu}_p)(\mathbf{x}_{bb} - \boldsymbol{\mu}_p)^\top] \tag{7}$$

$$\boldsymbol{\mu}_{q_\theta} := \mathbb{E}_{\mathbf{x} \sim q_\theta}[\mathbf{x}_{bb}], \quad \boldsymbol{\Sigma}_{q_\theta} := \mathbb{E}_{\mathbf{x} \sim q_\theta}[(\mathbf{x}_{bb} - \boldsymbol{\mu}_{q_\theta})(\mathbf{x}_{bb} - \boldsymbol{\mu}_{q_\theta})^\top] \tag{8}$$

are mean and covariance, respectively, of the vectorized backbone atom distance matrices.

(3) As a third stage of training, we train with a combination of the NLL, the 2-Wasserstein loss, and the KL divergence. In our final stage of training, we drop the 2-Wasserstein loss term and train to minimize a combination of the NLL and the KL divergence.

---

[1]We refer to the reverse KL divergence as just "KL divergence" or "KL loss", as is often done in the literature.

## 4 RESULTS

### 4.1 PROTEIN SYSTEMS

**Alanine dipeptide (ADP)** is a two residue (22-atoms) common benchmark system for evaluating BGs (Noé et al., 2019; Köhler et al., 2021; Midgley et al., 2022). We use the MD simulation datasets provided by Midgley et al. (2022) for training and validation.

**HP35**(nle-nle), a 35-residue double-mutant of the villin headpiece subdomain, is a well-studied structure whose folding dynamics have been observed and documented (Beauchamp et al., 2012). For training, we use the MD simulation dataset made publicly available by Beauchamp et al. (2012) and remove faulty trajectories and unfolded structures as done by Ichinomiya (2022).

**Protein G** is a 56-residue cell surface-associated protein from *Streptococcus* that binds to IgG with high affinity (Derrick & Wigley, 1994). In order to train our model, we generated samples by running a MD simulation. The crystal structure of protein G, 1PGA, was used as the seed structure. The conformational space of Protein G was first explored by simulations with ClustENMD (Kaynak et al., 2021). From 3 rounds of ClustENMD iteration and approximately 300 generated conformations, 5 distinctly diverse structures were selected as the starting point for equilibrium MD simulation by Amber. On each starting structure, 5 replica simulations were carried out in parallel with different random seeds for 400 ns at 300 K. The total simulation time of all the replicas was accumulated to 1 microsecond. Thus, $10^6$ structures of protein G were saved over all the MD trajectories.

As a baseline model for comparison, we use Neural Spline Flows (NSF) with 58 rational quadratic spline coupling layers (Durkan et al., 2019). NSFs have been used in many recent works on BGs (Köhler et al., 2022; Midgley et al., 2022; Mahmoud et al., 2022). In particular, Midgley et al. (2022) utilized the NSF model (with fewer coupling layers) in their experiments with alanine dipeptide, a two residue system. In our experiments with ADP and HP35, we utilize 48 GA-RQS coupling layers for the backbone followed by 10 GA-RQS coupling layers for the full latent size. We also ensure that all models have a similar number of trainable parameters. We use a Gaussian base distribution for non-dihedral coordinates. For ADP and HP35, we use a uniform distribution for dihedral coordinates. For protein G, we use a von Mises base distribution for dihedral coordinates; we noticed that using a von Mises base distribution improved training for the protein G system as compared to a uniform or truncated normal distribution.

### 4.2 MAIN RESULTS

From Table 1, we see that our model has marginal improvements over the baseline model for ADP. This is not surprising as the system is extremely small, and maximum likelihood training sufficiently models the conformational landscape.

For both proteins, our model closely captures the individual residue flexibility as analyzed by the root mean square fluctuations (RMSF) of the generated samples from the various training schemes in Fig. 3(a). This is a common metric for MD analysis, where larger per-residue values indicate larger movements of that residue relative to the rest of the protein. Fig. 3(a) indicates that our model generates samples that present with similar levels of per-residue flexibility as the training data.

Table 1 displays $\Delta D$, the mean energy, and the mean NLL of structures generated from flow models trained with different strategies. For each model (architecture and training strategy), we generate $3 \times 10^6$ conformations ($10^6$ structures over 3 random seeds) after training with either protein G or Villin HP35. Due to the cost of computing $\Delta D$, we compute it for batches of $10^3$ samples and report statistics (mean and standard deviation) over the $3 \times 10^3$ batches. Before we computed sample statistics for the energy $u$, we first filtered out the samples with energy higher than the median value. This was done to remove high energy outliers that are not of interest and would noise the reported mean and standard deviations. We also report the mean and standard deviation (across 3 seeds) for the average NLL. We see that our model is capable of generating low-energy, stable conformations for these two systems while the baseline method and ablated training strategies produce samples with energies that are positive and five or more orders of magnitude greater.

Table 1 highlights a key difference in the results for protein G and villin HP35. For villin, models trained by reverse KL and without the 2-Wasserstein loss do not result in completely unraveled

Table 1: **Training BGs with different strategies.** We compute $\Delta D$, energy $u(\cdot)$, and mean NLL of $10^6$ generated structures after training with different training strategies with ADP, protein G, and Villin HP35. $\Delta D$ is computed for batches of $10^3$ samples. Means and standard deviations are reported. Statistics for $u(\cdot)$ are reported for structures with energy below the median sample energy. Best results are bold-faced. For reference, the energy for training data structures is $-317.5 \pm 125.5$ kcal/mol for protein G and $-1215.5 \pm 222.2$ kcal/mol for villin HP35. We compare our results against a Neural Spline Flows (NSF) baseline model.

| System | Arch. | Training strategy | | | $\Delta D$ (Å) | Energy $u(\mathbf{x})$ (kcal/mol) | $-\mathbb{E}_{p(\mathbf{x})}[\log q_\theta(\mathbf{x})]$ |
| | | NLL | KL | W2 | | | |
|---|---|---|---|---|---|---|---|
| | NSF | ✓ | | | $0.09 \pm 0.01$ | $(-1.19 \pm 0.61) \times 10^1$ | $38.29 \pm 0.19$ |
| ADP | Ours | ✓ | | | $0.08 \pm 0.01$ | $(-1.18 \pm 0.65) \times 10^1$ | $\mathbf{36.15 \pm 0.15}$ |
| | | ✓ | ✓ | | $0.05 \pm 0.01$ | $(-1.20 \pm 0.59) \times 10^1$ | $38.66 \pm 0.19$ |
| | | ✓ | | ✓ | $\mathbf{0.04 \pm 0.00}$ | $(-1.06 \pm 0.74) \times 10^1$ | $38.12 \pm 0.03$ |
| | Ours | ✓ | ✓ | ✓ | $\mathbf{0.03 \pm 0.01}$ | $(-1.31 \pm 0.52) \times 10^1$ | $37.67 \pm 0.09$ |
| | NSF | ✓ | | | $2.92 \pm 0.80$ | $(2.15 \pm 3.31) \times 10^{10}$ | $-263.46 \pm 0.13$ |
| Protein G | Ours | ✓ | | | $1.81 \pm 0.14$ | $(9.47 \pm 15.4) \times 10^8$ | $\mathbf{-310.11 \pm 0.08}$ |
| | | ✓ | ✓ | | $16.09 \pm 1.14$ | $(2.86 \pm 0.62) \times 10^2$ | $-308.68 \pm 0.08$ |
| | | ✓ | | ✓ | $\mathbf{0.18 \pm 0.01}$ | $(2.68 \pm 4.31) \times 10^6$ | $-307.17 \pm 0.01$ |
| | Ours | ✓ | ✓ | ✓ | $\mathbf{0.19 \pm 0.01}$ | $(-3.04 \pm 1.24) \times 10^2$ | $-309.10 \pm 0.91$ |
| | NSF | ✓ | | | $0.81 \pm 0.06$ | $(7.78 \pm 17.4) \times 10^7$ | $687.95 \pm 1.92$ |
| HP35 | Ours | ✓ | | | $0.65 \pm 0.04$ | $(5.29 \pm 11.7) \times 10^6$ | $\mathbf{651.90 \pm 2.88}$ |
| | | ✓ | ✓ | | $0.61 \pm 0.04$ | $(6.46 \pm 14.3) \times 10^2$ | $678.38 \pm 0.87$ |
| | | ✓ | | ✓ | $\mathbf{0.38 \pm 0.03}$ | $(1.15 \pm 1.76) \times 10^7$ | $678.31 \pm 1.55$ |
| | Ours | ✓ | ✓ | ✓ | $\mathbf{0.39 \pm 0.03}$ | $(-4.66 \pm 3.52) \times 10^2$ | $667.45 \pm 2.04$ |

structures. This is consistent with the notion that long-range interactions become much more important in larger structures. From Fig. 3(b), we see that Villin HP35 is not densely packed, and local interactions, e.g., as seen in $\alpha$-helices, are more prevalent than long-range interactions/contacts. In addition, we see that our model generates diverse alternative modes of the folded villin HP35 protein that are energetically stable compared to the structures obtained from the baseline model.

Fig. 3(d) visualizes pathological structures of protein G generated via different training schemes. In Fig. 3(d, left), we see that minimizing the NLL generally captures local structural motifs, such as the $\alpha$-helix. However, structures generated by training only with the NLL loss tend to have clashes in the backbone, as highlighted with red circles in Fig 3(d), and/or long range distortions. This results in large van der waals repulsion as evidenced by the high average energy values in Table 1.

In Fig. 3(d, middle), we see that structures generated by minimizing a combination of the NLL loss and the reverse KL divergence unravel and present with large global distortions. This results from large, perpetually unstable gradients during training. In Fig. 3(d, right), we see that training with a combination of the NLL loss and the 2-Wasserstein loss properly captures the backbone structural distribution, but tends to have clashes in the sidechains. Table 1 demonstrates that only our model with our multistage training strategy is able to achieve both low energy samples and proper global structures. The 2-Wasserstein loss prevents large backbone distortions, and thus, simultaneously minimizing the reverse KL divergence accelerates learning for the side chain marginals with respect to the backbone and other side chain atoms.

### 4.3 BGs CAN GENERATE NOVEL SAMPLES

One of the primary goals for development of BG models is to sample important metastable states that are unseen or difficult to sample by conventional MD simulations. Protein G is a medium-size protein with diverse metastable states that provide a system for us to evaluate the capability of our

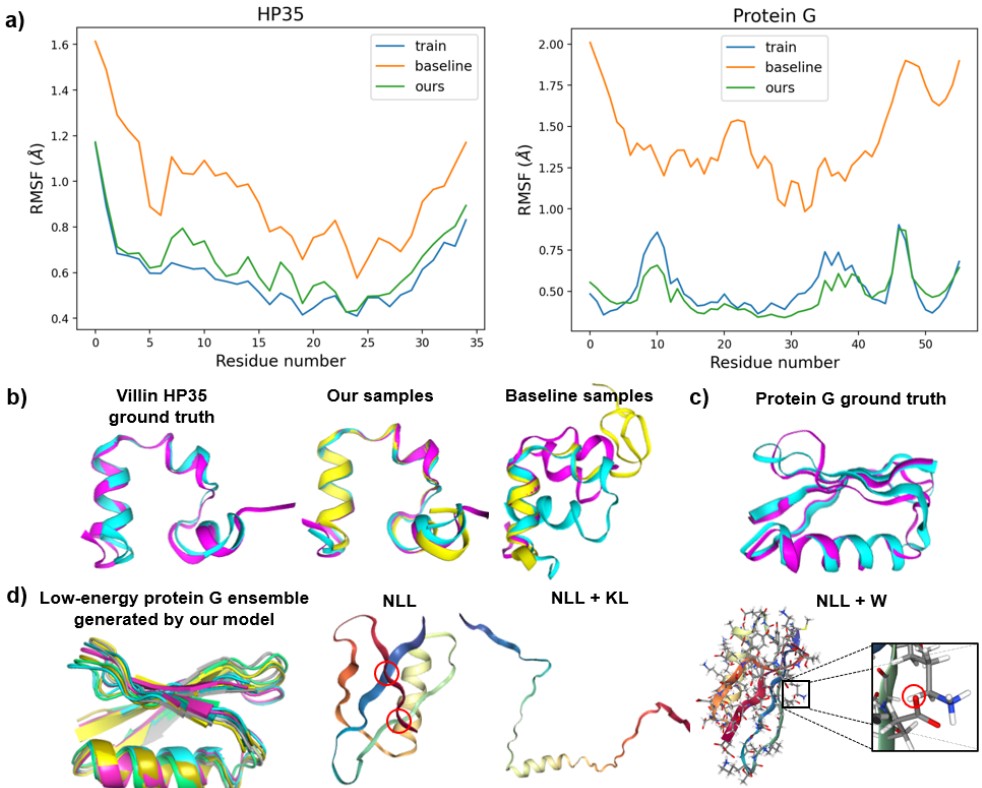

Figure 3: **Sample conformations generated by BG via different training strategies.** (a) Root mean square fluctuation (RMSF) computed for each residue (C$\alpha$ atoms) in HP35 and protein G. Matching the training dataset's plot is desirable. (b) Examples of HP35 from ground truth training data, generated samples from our model, and generated samples from the baseline model. (c) Example of two metastable states from protein G training data. (d) Low-energy conformations of protein G generated by our model superimposed on each other. We also show some examples of pathological structures generated after training with different training paradigms: NLL (maximum likelihood), both NLL and KL divergence, and NLL and the 2-Wasserstein loss. Atom clashes are highlighted with red circles.

BG model. First, we visualize 2D UMAP embeddings (McInnes et al., 2018) for the training data set, test dataset, and for $2 \times 10^5$ generated samples of protein G in Fig. 4(a). We see that the test dataset 4(a, middle), an independent MD dataset, covers far less conformational space than an equivalent number of BG samples as shown in Fig. 4(a, right).

Secondly, we computed, respectively, the energy distributions of the training set from MD simulations and sample set from the BG model as shown by Fig. 4(c). Unlike the training set, the BG sample energy distribution is bimodal. Analysis of structures in the second peak revealed a set of conformations not present in the training set. These new structures are characterized by a large bent conformation in the hair-pin loop which links beta-strand 3 and 4 of protein G. Fig. 4(b) compares representative structures of the discovered new structure (magenta) with the closest structure (by RMSD) in the training dataset (cyan). We also see vastly different sidechain conformations along the bent loops between two structures. Energy minimization on the discovered new structures demonstrated that these new structures are local-minimum metastable conformations. Thirdly, we carefully examined the lowest-energy conformations generated by the BG model. Fig. 4(d) showcases a group of lowest-energy structures generated by the BG model, overlaid by backbone and all-atom side chains shown explicitly. All of these structures are very similar to the crystal structure of protein G, demonstrating that the trained BG model is capable of generating protein structures with high quality at the atomic level.

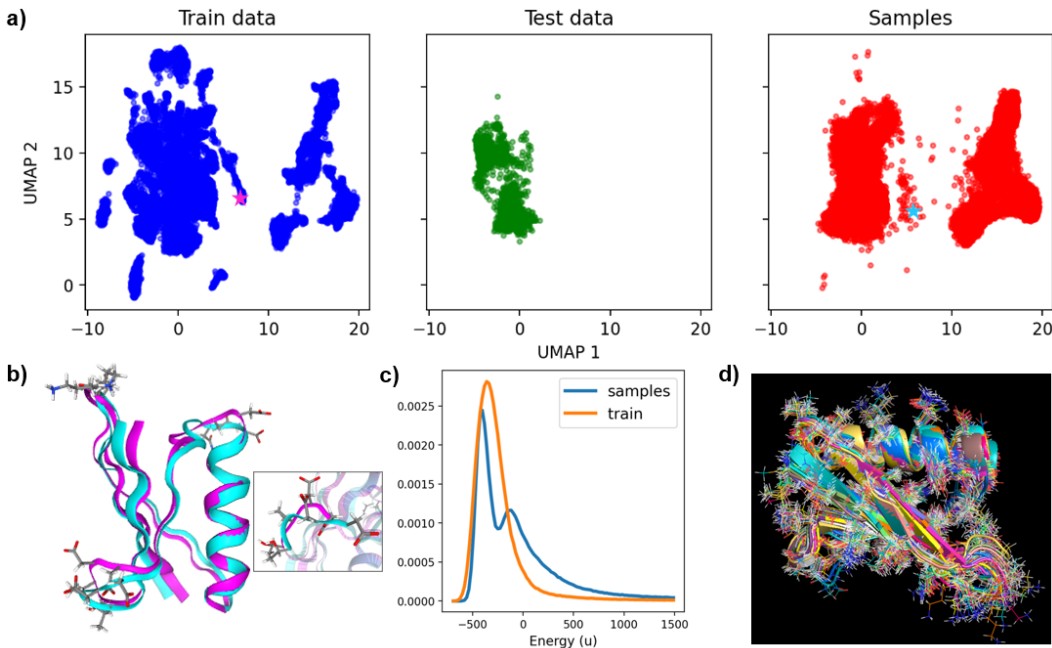

Figure 4: **BGs can generate novel sample conformations.** (a) Protein G 2D UMAP embeddings for the training data, test data, and $2 \times 10^5$ generated samples. (b) A representative example of generated structures by the BG model which was not found in training data (cyan) and the closest structure in the training dataset (magenta) by RMSD. Both structures are depicted as stars with their respective structural colors in (a). (c) Protein G energy distribution of training dataset (orange) and samples (blue) generated by our model. The second energy peak of the sampled conformations covers the novel structure shown in (b). (d) An overlay of high-resolution, lowest-energy all-atom structures of protein G generated by the BG model. This demonstrates that our model is capable of sampling low-energy conformations at atomic resolution.

## 5 DISCUSSION

The scalability of the Boltzmann generator for large macromolecules is a major challenge. This study developed a new flow model architecture to address this challenge. We represented protein structures using internal coordinates and conducted conformational sampling primarily in dihedral angle space. This approach reduces the number of variables to be trained in the flow model, and conformational sampling follows the energetics of proteins. We made several innovations to the neural network architecture, such as using gated attention units for transformation maps and rotary positional embedding to capture non-local interactions. We also introduced split channels to allocate more transformation layers for backbone representations and employed a 2-Wasserstein loss with respect to distance matrices of the backbone atoms to consider long-distance interactions. We demonstrated the feasibility of this new flow model by successfully training it on medium-sized proteins. The model generated interesting results by sampling new metastable states that are difficult to obtain through conventional MD simulations.

We envision further improvement to scalability may be possible via various means. The success of the 2-Wasserstein loss motivates further exploration of spatial loss functions such as using distance matrices or contact maps. Li et al. (2023) showed that the native folding conformations of proteins can be more accurately sampled when backbone bond angles are conditioned on backbone dihedral angles, which could further simplify our representation. One primary limitation of our work is the lack of transferability between molecular systems. Another limitation is that, as a normalizing flow model, the model has a large number of parameters. Conditional diffusion-based approaches are a promising direction to address these limitations.

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
