# A  Related works

## I.  Boltzmann Generators

Bolzmann generators [Noé et al., 2019] are normalizing flows that approximate Boltzmann distributions. Noé et al. [2019] utilized the fact that normalizing flows are tractable density models and introduced a notion of training by energy via reverse KL-divergence minimization. Recently, there has been a growing interest in Boltzmann generators. Dibak et al. [2022] proposed temperature steerable flows that generalized to families of ensembles across multiple temperatures, thereby increasing the range of thermodynamic states accessible for sampling. Unfortunately, this model tends to undersample significant local minimas for systems as small as alanine dipeptide. The authors believed that this was due to the limited expressivity of the flow model. Wu et al. [2020] proposed stochastic normalizing flows, which combine flows with MCMC methods by introducing sampling layers between flow layers to improve model expressivity. Unfortunately, this method is computationally expensive as it involves many more target evaluations. In addition, stochastic normalizing flows tend to miss modes [Midgley et al., 2022]. Köhler et al. [2021] introduced smooth normalizing flows, which are $C^\infty$-smooth, thus making them more physically amenable. They also introduce force-matching as an added loss term. While they have impressive results and modal coverage for alanine dipeptide, they utilize a root-finding algorithm to approximate the inverse for their smooth flows, which becomes computationally prohibitive for higher-dimensional systems.

This work has focused on normalizing flows. However, diffusion models have also shown great promise as an alternative generative model for learning Boltzmann generators. Jing et al. [2022] train a diffusion model to learn the Boltzmann distribution over the torsion angles of multiple drug-like molecules, while using cheminformatics methods for the bond lengths and angles. They perform energy-based training (similar in spirit to the reverse KL divergence in flow model training) via estimation of a score matching loss using samples generated by the model. However, this method does not scale well to larger molecules and inherits the same problem of unstable training at initialization.

## II.  Loss functions

Wirnsberger et al. [2022] trained a flow model without MD samples by minimizing the KL divergence to approximate the Boltzmann distribution of atomic solids with up to 512 atoms. However, the KL-divergence suffers from mode-seeking behavior, which severely impairs training for multimodal target distributions. While the forward KL-divergence, i.e. maximum likelihood, is mass covering, the Monte Carlo approximations of such an objective have a very high variance in loss. To circumvent this, Midgley et al. [2022] trains a flow to approximate a target $p$ by minimizing the alpha-divergence with $\alpha = 2$, which is estimated with annealed importance sampling (AIS) using the flow $q$ as the base distribution and $p^2/q$ as target. This method is notable in that it does not require any MD samples but still achieves impressive results for alanine dipeptide. Nonetheless, the AIS component is computationally expensive and scales poorly for larger systems.

## III.  Coarse-graining

Several works have attempted to scale flow-based Boltzmann generators to larger systems. Mahmoud et al. [2022] trained a flow model on coarse-grained protein representations which they then mapped back to full-atom representations using a language model. On a similar note, Köhler et al. [2022] trained a normalizing flow to represent the probability density for coarse-grained (CG) MD samples in order to learn the parameters of a CG model. Unfortunately, coarse-grain approaches tend to lose significant information compared to full-atom resolution for downstream applications. Importantly, both works note that using internal-coordinate representations do not scale well as small changes in torsion angles can lead to large global distortions. Our results indicate that this is not necessarily true, as we use a reduced internal-coordinate representation.

## IV.  Normalizing flow architectures

Our flow model, while novel, shares some similarities to previous works. DenseFlow [Grcić et al., 2021] fuses a densely connected convolutional block with Nyström self-attention in modules with both cross-unit and intra-module couplings. This architecture is specifically designed for image

data and utilizes a linear approximation for the self-attention mechanism. In contrast, we use gated attention and rotary positional embeddings in order to handle the sequential nature of proteins.

Multiscale flow architectures were first introduced by Dinh et al. [2017] In the protein domain, previous works also split the inputs into different channels [Noé et al., 2019, Köhler et al., 2021, 2022]. However, they split the input dimensions into torsion, angle, and bond channels. In contrast, our model splits the input into separate backbone and sidechain channels to better capture the global distribution.

## V.  Transferable models

As mentioned in the discussion section of the main text, one of the primary limitations of this work is the inability to transfer across molecular systems. Several works have attempted to overcome this limitation. Klein et al. [2023a] developed Timewarp: an enhanced sampling method which uses a normalising flow as a proposal distribution in a Markov Chain Monte Carlo (MCMC) method targeting the Boltzmann distribution. However, the transferability of Timewarp is demonstrated only for small peptides (2-4 amino acids), and its capabilities are yet to be validated on larger systems. One promising direction for developing transferable models targeting the Boltzmann distribution is diffusion modeling. Jing et al. [2022] develop a torsion score model that allows for transferability across systems. However, their model is only trained and validated on small, drug-like molecules that are around the same size as alanine dipeptide or smaller. Fu et al. [2023] trained a multi-scale graph neural network that directly simulates coarse-grained MD with a very large time step and used a diffusion model as a refinement module to mitigate simulation instability. The degree of coarse-graining as presented in the paper diminishes the resolution, thereby making downstream drug-design applications infeasible. In addition, coarse-graining dynamics often do not mimic real transitions that occur in nature for proteins.

## VI.  Equivariant flow models

Several recent works have attempted to bring the benefits of equivariance (particularly SE(3) equivariance) [Thomas et al., 2018, Kondor and Trivedi, 2018a,b] to normalizing flow models. Two recent works, in particular, were able to model the Boltzmann distribution for alanine dipeptide in Cartesian coordinates. Midgley et al. [2023] develop an augmented coupling flow that preserve SE(3) and permutation equivariance that can sample from the Boltzmann distribution of alanine dipeptide via importance weighting. Klein et al. [2023b] utilize a different generative modeling method called flow matching. Specifically, they utilize equivariant flow matching to exploit the physical symmetries of the Boltzmann distribution and achieve significant sampling efficiency for alanine dipeptide. Unfortunately, both works still fall short of internal coordinate-based methods for alanine dipeptide. However, as a Cartesian coordinate representation is more generalizable and will often present with smoother gradients and more stable training, they are a promising direction for developing scalable BGs.

## B   Training by energy

Below, we show the connection between minimizing the reverse KL-divergence and minimizing the energy of generated samples.

$$
\begin{aligned}
KL(q_\theta || p) &= \mathbb{E}_{\mathbf{x} \sim q_\theta} \left[ \log q_\theta(\mathbf{x}) - \log p(\mathbf{x}) \right] \\
&= \mathbb{E}_{\mathbf{z} \sim q} \left[ \log q(\mathbf{z}) - \log |\det(J_{f_\theta}(\mathbf{z}))| - \log p(f_\theta(\mathbf{z})) \right] \\
&= -H_{\mathbf{z}} + \log C + \mathbb{E}_{\mathbf{z} \sim q} \left[ u(f_\theta(\mathbf{z})) - \log |\det(J_{f_\theta}(\mathbf{z}))| \right],
\end{aligned}
$$

where $H_{\mathbf{z}}$ is the entropy of the random variable $\mathbf{z}$ and $C = \int e^{-u(\mathbf{x})/(kT)} d\mathbf{x}$ is the normalization constant for the Boltzmann distribution $p(\mathbf{x}) \propto e^{-u(\mathbf{x})/(kT)}$. When minimizing the KL-divergence with respect to the parameters $\theta$, the entropy term and the log normalization constant disappear as they are not dependent on $\theta$:

$$
\begin{aligned}
\theta^* &= \operatorname*{argmin}_{\theta} -\cancel{H_{\mathbf{z}}} + \cancel{\log C} + \mathbb{E}_{\mathbf{z} \sim q} \left[ u(f_\theta(\mathbf{z})) - \log |\det(J_{f_\theta}(\mathbf{z}))| \right] \\
&= \operatorname*{argmin}_{\theta} \mathbb{E}_{\mathbf{z} \sim q} \left[ u(f_\theta(\mathbf{z})) - \log |\det(J_{f_\theta}(\mathbf{z}))| \right].
\end{aligned}
$$

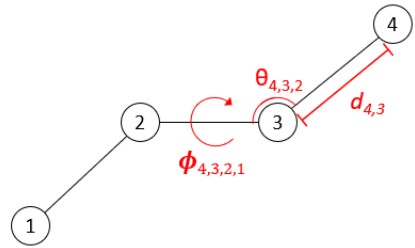

Fig. S.1: An illustration of the definition of bond length, bond angle, and dihedral angle by four atoms. Subscripts indicate the atoms that define the value, where order is given by the bond graph connectivity. In internal coordinate system, the position or Cartesian coordinate of atom 4 is determined by atom 1,2 and 3 based on bond length, bond angle and dihedral angle.

The expectation here is usually approximated with a Monte Carlo estimate, but a variety of different sampling procedures can be utilized. The log determinant Jacobian (ldj) term can be seen as promoting entropy, or exploration, of the sample space.

## C  Coordinate Transformation

### I.  Protein Structure

Protein structure refers to the three-dimensional arrangement of atoms in an amino acid-chain molecule. There are four distinct levels by which we can describe protein structure. The *primary structure* of a protein refers to the sequence of amino acids in the polypeptide chain. The *secondary structure* refers to regularly patterned local sub-structures on the actual polypeptide backbone chain. The two most common secondary structure motifs are $\alpha$-helices and $\beta$-sheets. Tertiary structure refers to the overall three-dimensional structure created by a single polypeptide. Tertiary structure is primarily driven by non-specific hydrophobic interactions as well as long-range intramolecular forces. Quaternary structure refers to the three-dimensional structure consisting of two or more polypeptide chains that operate as a single functional unit.

### II.  Coordinate Representations

Boltzmann generators usually do not operate directly with Cartesian coordinates. The primary global conformational changes of a protein do not described efficiently by the atomic Cartesian coordinates. This is driven by the fact that chemical bonds are very stiff, and energetically-favored conformational changes take place via rotations around single chemical bonds [Vaidehi and Jain, 2015]. A more commonly used alternative is internal coordinates. Internal coordinates are defined by bond lengths $d$, bond angles $\theta$, and dihedral angles $\phi$ (Fig. S.1).

In their seminal work, Noé et al. [2019] introduced a coordinate transformation whereby the protein backbone atoms (primarily defined as the $N$, $C_\alpha$, and $C$ atoms) are mapped PCA coordinates while the rest of the atoms are mapped to internal coordinates. The motivation behind this mixed coordinate transformation is that protein conformations are highly sensitive to changes in backbone internal coordinates. This often results in unstable training and difficulty in generating natural, i.e., high Boltzmann probability, structures. Most works since have used full internal coordinate representations but experimented only with small systems, the most common of which is alanine dipeptide (22 atoms). Köhler et al. [2022] note that scaling Boltzmann generators to larger systems is difficult with internal coordinate representations.

In our work, rather than using a full internal coordinate representation, which would be $3N - 6$ dimensional (where $N$ is the number of atoms in the system), we utilize a reduced internal coordinate representation. For training features, we use the dihedral angles and the bond angles for the 3 backbone atoms ($N, C_\alpha, C$). For side-chain atoms, we use all rotatable dihedral angles around single bond. All bond lengths and bond angles other than the 3 defining backbone atoms and improper torsion angles are kept at their mean values calculated from input protein structures. By examining all protein structures generated, we confirmed that such a reduced internal coordinate system can

Table S.1: Hyperparameters for training

| | |
|---|---|
| Optimizer | AdamW |
| $\lambda$ | 0.0001 |
| Learning rate | 0.002 |
| Scheduler | ReduceLROnPlateau |
| Patience (epochs) | 5 |
| Factor | 0.1 |
| Batch Size | 256 |
| Dropout | 0.1 |
| $Q_{\text{dim}}$ | 32 |
| $K_{\text{dim}}$ | 32 |
| $V_{\text{dim}}$ | 64 |
| Normalization | Scaled |
| Attention | Laplace |
| Activation | SiLU |
| RQS bins | 8 |
| Epochs (NLL) | 200 |
| Epochs (NLL + W) | 50 |
| Epochs (NLL + W + KL) | 20 |
| Epochs (NLL + KL) | 10 |

represent all protein structures to very high accuracy and quality. Recent works have adopted similar approaches; Wu et al. [2022] utilize only the backbone torsion and bond angles to represent various proteins, while Wang et al. [2022] simply use the backbone torsion angles to represent the polypeptide AiB9.

## D  Training details and architecture

All models were trained on a single NVIDIA A100 GPUs with the Adam optimizer and a dropout factor of 0.1. For model that utilized GAU-RQS blocks, the dimensionalities of the $Q$, $K$, and $V$ matrices were 32, 32, and 64, respectively. In addition, we utilized scaled normalization [Nguyen and Salazar, 2019], the Laplace attention function [Ma et al., 2023], and SiLU activations [Ramachandran et al., 2017]. For the gated attention units, we also use the T5 relative positional bias [Raffel et al., 2020]. For the rational quadratic splines (RQS), we use a bin size of $K = 8$.

Data was all standard normalized. Dihedral angles were constrained to be within $[-\pi, \pi]$ and shifted as done by Sittel et al. [2017].

For the multi-stage training strategy, all models were trained for 200 epochs ( 12 hours) with the NLL loss, 50 epochs ( 8 hours) with NLL+W, 20 epochs ( 8 hours) with NLL+W+KL, and 10 epochs ( 3 hours) with NLL+KL. The approximate times are for protein G, which has 56 residues.

We do no hyperparameter tuning due to the lack of compute and time. Further implementation details are given in the code, which is available upon request. A summary of the hyperparameters for our model are provided in Table S.1.

## E  Further ablations

In this section, we provide further ablations for Table 1 in the main text. In particular, we provide further ablations with regards to the training strategy with the baseline neural spline flows (NSF) architecture in Table S.2. As we can see from the table, while our different training strategies improve upon the baseline model with NLL training, the improvements are not as drastic as for our architecture (Table 1).

Table S.2: **Training NSF basline model with different strategies.** We compute $\Delta D$, energy $u(\cdot)$, and mean NLL of $10^6$ generated structures after training with different training strategies for the baseline NSF model with ADP, protein G, and Villin HP35.

| System | NLL | Training strategy KL | W2 | $\Delta D$ (Å) | Energy $u(\mathbf{x})$ (kcal/mol) | $-\mathbb{E}_{p(\mathbf{x})}[\log q_\theta(\mathbf{x})]$ |
|---|---|---|---|---|---|---|
| ADP | ✓ | | | $0.09 \pm 0.01$ | $(-1.19 \pm 0.61) \times 10^1$ | $38.29 \pm 0.19$ |
| | ✓ | ✓ | | $0.08 \pm 0.01$ | $(-1.21 \pm 0.48) \times 10^1$ | $40.11 \pm 0.20$ |
| | ✓ | | ✓ | $0.05 \pm 0.01$ | $(-0.99 \pm 0.57) \times 10^1$ | $41.03 \pm 0.08$ |
| | ✓ | ✓ | ✓ | $0.04 \pm 0.00$ | $(-1.22 \pm 0.13) \times 10^1$ | $39.10 \pm 0.13$ |
| Protein G | ✓ | | | $2.92 \pm 0.80$ | $(2.15 \pm 3.31) \times 10^{10}$ | $-263.46 \pm 0.13$ |
| | ✓ | ✓ | | $18.19 \pm 2.88$ | $(2.90 \pm 0.82) \times 10^2$ | $-260.87 \pm 0.51$ |
| | ✓ | | ✓ | $1.81 \pm 0.33$ | $(6.04 \pm 3.79) \times 10^7$ | $-261.01 \pm 0.33$ |
| | ✓ | ✓ | ✓ | $1.58 \pm 0.29$ | $(-0.86 \pm 2.04) \times 10^2$ | $-257.82 \pm 0.92$ |
| HP35 | ✓ | | | $0.81 \pm 0.06$ | $(7.78 \pm 17.4) \times 10^7$ | $687.95 \pm 1.92$ |
| | ✓ | ✓ | | $0.91 \pm 0.05$ | $(2.15 \pm 11.4) \times 10^3$ | $691.41 \pm 1.47$ |
| | ✓ | | ✓ | $0.59 \pm 0.05$ | $(9.61 \pm 2.55) \times 10^7$ | $690.87 \pm 1.05$ |
| | ✓ | ✓ | ✓ | $0.61 \pm 0.07$ | $(-1.77 \pm 1.49) \times 10^2$ | $691.10 \pm 2.12$ |

Table S.3: Effective sample size (ESS) of the various training strategies and architectures.

| System | Arch. | Training strategy NLL | KL | W2. | ESS (%) |
|---|---|---|---|---|---|
| ADP | NSF | ✓ | | | $3.9 \pm 0.5$ |
| | Ours | ✓ | | | $9.1 \pm 1.7$ |
| | | ✓ | ✓ | | $58.6 \pm 8.4$ |
| | | ✓ | | ✓ | $39.4 \pm 2.7$ |
| | Ours | ✓ | ✓ | ✓ | $\mathbf{88.4 \pm 0.2}$ |
| Protein G | NSF | ✓ | | | $0.0 \pm 0.0$ |
| | Ours | ✓ | | | $0.0 \pm 0.0$ |
| | | ✓ | ✓ | | $0.0 \pm 0.0$ |
| | | ✓ | | ✓ | $0.0 \pm 0.0$ |
| | Ours | ✓ | ✓ | ✓ | $\mathbf{62.47 \pm 1.4}$ |
| HP35 | NSF | ✓ | | | $0.0 \pm 0.0$ |
| | Ours | ✓ | | | $0.0 \pm 0.0$ |
| | | ✓ | ✓ | | $0.0 \pm 0.0$ |
| | | ✓ | | ✓ | $0.0 \pm 0.0$ |
| | Ours | ✓ | ✓ | ✓ | $\mathbf{43.9 \pm 1.3}$ |

## F    Reweighted distribution

Typically, the output distribution of the flow model will not match exactly with the target distribution, and previous works employ importance sample reweighting to the target distribution [Noé et al., 2019, Midgley et al., 2022, Wu et al., 2020]. While efficient Boltzmann reweighting is feasible for a small system like alanine dipeptide, the current work makes several modeling assumptions to scale to larger molecules. We model a distribution on a space with reduced dimensionality, which is not exactly the Boltzmann distribution. To be precise, we model $p(\tau, \theta_{bb}|L = \bar{L})$, where $\tau$ are torsion angles, $\theta_{bb}$ denotes backbone bond angles, and $L$ and $\bar{L}$ denote other internal coordinates and their mean marginals, respectively. In addition, while the MD simulations for the training data were trained with explicit water, we use an implicit water model (for efficiency) when training with the energy function. This results in a wider range of energy values for our training data and generated samples for protein

167 G and villin HP35 (the alanine dipeptide data is open source and is run with an implicit solvent). Due
168 to precision issues, it is difficult to meaningfully compare importance weights as only the lowest
169 energy structures will tend to have a nonzero importance weight. For these reasons, we consider
170 the histogram of our training data distribution for $e^{-u(\mathbf{x})}$, and use the bins and densities to define
171 $p_{data}(\mathbf{x})$, our target distribution. We define $q(\mathbf{x})$ as the likelihood according to our flow model.

172 We report the effective sample size (ESS) [Martino et al., 2017] (Table S.3) and display the reweighted
173 energy distribution according to the energies computed for the training data distribution of protein G
174 and villin HP35 (Fig. S.2). As we can see from the table, for larger systems such as protein G and
175 HP35, only the model that utilizes our novel architecture and multi-stage training strategy are capable
176 of capturing a meaningful subset of the data distribution, as measured by ESS. This is primarily due
177 to the fact that the other models generate samples with atomic clashes that dramatically increase their
178 associated energies.

179 To remedy some of the issues with using the Boltzmann distribution as our target distribution, we
180 computed the energies of the generated structures and the training data with force field parameters
181 that more closely modeled the simulating force field (specifically, we use the GBn2 implicit solvent
182 model). We then set the target distribution as $p(\mathbf{x}) \propto e^{-u(\mathbf{x})}$. We conduct importance sampled
183 reweighting and display the results for protein G in Fig S.3. As we can see, Boltzmann reweighting
184 tends to sample only for the lowest energy states. In fact, samples from the training data would rarely
185 have nonzero weights. This motivated our previous approach for Fig. S.2.

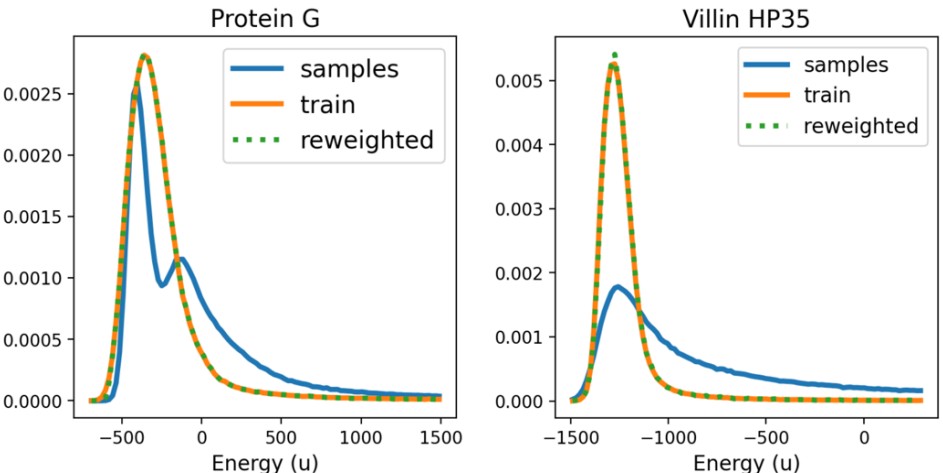

Fig. S.2: Energy distribution of the training data (orange), samples generated from the model (blue),
and the importance weight resampled energy distribution from the flow model (green) for protein G
and HP35. The target distribution for importance weighting is set as the histogram distribution of
$p_{data}(\mathbf{x}) \propto e^{-u(\mathbf{x})}$. Energy here is computed in a vacuum.

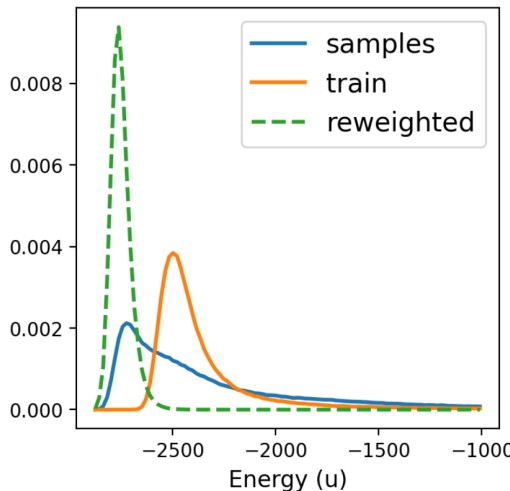

Fig. S.3: Energy distribution of the training data (orange), samples generated from the model (blue), and the importance weight resampled energy distribution from the flow model (green) for protein G. We modify the force field for computing the energies to be closer to the simulating force field. The target distribution for importance weighting is the Boltzmann probability $p(\mathbf{x}) \propto e^{-u(\mathbf{x})}$.