# OpenReview forum: "Scalable Normalizing Flows Enable Boltzmann Generators for Macromolecules"
_ICLR.cc/2024/Conference — Submitted to ICLR 2024_

### Official Review · Reviewer_3BfG · 2023-10-30

**Soundness:** 2 fair
**Presentation:** 3 good
**Contribution:** 3 good
**Rating:** 6
**Confidence:** 4

**Summary:**

The authors introduce a new flow architecture and training loss to train Boltzmann generators for large proteins. The flow architecture operates on internal coordinates of the backbone and the side chains, while bonds and bond angles of the side chain remain fixed. In addition to the commonly used forward and reverse KL divergence to train Boltzmann generators, they also employ a novel loss based on the distribution of the backbone all atom distance matrix.
They show in their experiments on protein G and HP35 that their Boltzmann generator is capable of generating samples with low energies. Moreover, they show for protein G that their Boltzmann generator samples meta-stable states unseen during training.

**Strengths:**

- The paper is well written and easy to follow.
- The authors introduce a new normalizing flow architecture, which uses preexisting building blocks.
- The main novelty is the training with the 2-Wasserstein distance loss that measures the derivation of the all distance matrix for the backbone between Boltzmann generator samples and samples from the target distribution.
- The new loss (in combination with the architecture) provides a way to scale Boltzmann generators to larger proteins
- Their experiments demonstrate that these trained Boltzmann generators produce samples with low energies, and in the case of protein G, they even generate structures that were previously unseen during the training phase.
- That the Boltzmann generators samples unseen states might allow to have non equilibrium training data that even misses some meta-stable states, which is important to gain advantages over traditional MD.
- The work hast the potential to be an important contribution for the community.

**Weaknesses:**

- One of the main goals of Boltzmann generators is to generate samples from the equilibrium Boltzmann distribution. As the output distribution will usually differ from the target distribution, Boltzmann generators employ reweighting to the target distribution (Noe et al. 2019). This allows to generate unbiased samples from the target distribution with the Boltzmann generator. However, efficiently doing reweighting requires that the output distribution is close to the target distribution. This can be measured using Kish's effective sample size (ESS), using the reweighting weights. Hence, the ESS should be reported in the paper. Merely presenting the mean energy values of samples does not provide a comprehensive assessment of the Boltzmann generator's ability to efficiently generate samples from the equilibrium distribution. In addition, also reweighted energies distribution could be shown.
- Boltzmann generators generally struggle to achieve a speed-up over traditional MD simulations, as they require data from these MD simulations to be trained.  The comparison presented in Figure 4 may not be entirely fair, given that generating Boltzmann generator samples necessitates the entire training dataset, which probably took considerably more time to generate than the test data.
- In general, it does not seem that the method could be made transferable, which makes it difficult to provide a real alternative to traditional MD simulations.
- The authors should consider discussing this limitation in their work, along with citing relevant work on transferable models for MD acceleration, such as references [1, 2, 3], and the recent advancements in Boltzmann generators for molecules in Cartesian coordinates, e.g. [4, 5]. These could be directly trained in a transferable manner, unlike Boltzmann generators operating in internal coordinates.
- The code is currently not available

In summary, I believe that if the authors address these concerns, including reporting the ESS and discussing limitations, their paper has the potential to make a substantial and impactful contribution to the community.

[1] Bowen Jing et al. Torsional diffusion for molecular conformer generation. NeurIPS, 2022

[2] Leon Klein et al. Timewarp: Transferable acceleration of molecular dynamics by learning time-coarsened dynamics. arXiv preprint arXiv:2302.01170, 2023

[3] Xiang Fu et al. Simulate timeintegrated coarse-grained molecular dynamics with multi-scale graph networks. Transactions on Machine Learning Research, 2023.

[4] Leon Klein et al. Equivariant flow matching. arXiv preprint arXiv:2306.15030, 2023.

[5] Laurence I Midgley et al. Se (3) equivariant augmented coupling flows. arXiv preprint arXiv:2308.10364, 2023

**Questions:**

- The 2-Wasserstin loss is supposed to measure the distance between the backbone atom all distance distributions. However, from the notation in Equation (6) it seems as they measure the distance of the backbone bond and torsion angles.
- Only the mean energy is reported. Although this is computed with only samples with below median energy, I suspect that this mean might still be distorted significantly. Could the authors comment on how the energy distribution looks like when all high energy samples are discarded? In general, high energy samples will have nearly zero weight and are therefore not relevant if the rest of the samples have reasonable energies. Alternatively, one might consider reporting the minimal sampled energy in addition to the mean. That way, it is easier to see which methods do not generate any useful samples.
- Can the authors elaborate on the time requirements for equilibrium Molecular Dynamics (MD) simulations in comparison to the time required for sampling with a Boltzmann generator?
- Have the authors attempted to train the baseline model using their proposed training method? This experiment could help discern the relative impact of the new loss function and the proposed architecture on the improved results.
- What does the energy distribution for HP35 look like? Is it close to the target distribution? What does the reweighted distribution look like?

---

> ### Author Response · Authors · 2023-11-19
>
> 1. > … the ESS should be reported in the paper … In addition, also reweighted energies distribution could be shown…
>
>     Thank you for your suggestion. While efficient Boltzmann reweighting is feasible for a small system like alanine dipeptide, the current work makes several modeling assumptions to scale to larger molecules. We model a distribution on a space with reduced dimensionality, which is not exactly the Boltzmann distribution. To be precise, we model $p(\mathbf{\tau}, \mathbf{\theta}_{bb} | L = \bar{L})$, where $\tau$ are torsion angles, $\theta\_{bb}$ denotes backbone bond angles, and $L$ and $\bar{L}$ denote other internal coordinates and their mean marginals, respectively. In addition, while the MD simulations for the training data were trained with explicit water, we use an implicit water model (for efficiency) when training with the energy function. This results in a wider range of energy values for our training data and generated samples for protein G and villin HP35 (the alanine dipeptide data is open source and is run with an implicit solvent). Due to precision issues, it is difficult to meaningfully compare importance weights.
>
>     Thus, we report the effective sample size (ESS) (Table S.3) and display the reweighted energy distribution according to the energies computed for the training data distribution of protein G (Fig. S.2). As we can see from the table, for larger systems such as protein G and HP35, only the model that utilizes our novel architecture and multi-stage training strategy are capable of capturing a meaningful subset of the data distribution, as measured by ESS. This is primarily due to the fact that the other models generate samples with atomic clashes that dramatically increase their associated energies.
>
> 2. > The comparison presented in Figure 4 may not be entirely fair, given that generating Boltzmann generator samples necessitates the entire training dataset, which probably took considerably more time to generate than the test data.
>
>     Indeed, training the Boltzmann generator requires simulation data. However, we note in Sec 4.3 and Fig 4 that our model is capable of generating novel conformations that are not found in the simulation data. In fact, the representative structure highlighted in Fig 4b involves a largely bent hairpin loop that is not found in multiple-seeded MD simulations despite its low energy. In addition, the test data simulation took just under a week to generate while our model generated one million samples in 20 minutes.
>
> 3. > The authors should consider discussing this limitation in their work, along with citing relevant work on transferable models for MD acceleration
>
>     Thank you for your suggestion. We have added some limitations of our model in the Discussion section. We have also added sections discussing transferable models for MD acceleration and recent advances in Cartesian coordinates for BGs in the supplements (Sec A V and A VI).
>
> 4. > The code is currently not available
>
> The code will be made available upon acceptance.
>
> 5. > However, from the notation in Equation (6) it seems as they measure the distance of the backbone bond and torsion angles
>
> In the line after equation (8), we specify that equations (7) and (8) are mean and covariance of the vectorized backbone atom distance matrices.
>
> 6. > Could the authors comment on how the energy distribution looks like when all high energy samples are discarded?
>
> This is a good point and we thank you for your suggestion. In the caption of Table 1, we note that statistics for the energy are reported after filtering out samples with energy higher than the median. We also took your suggestion in point (1), and in Fig S.2, we present a re-weighted energy distribution for the samples generated for protein G and villin HP35.
>
> 7. > Can the authors elaborate on the time requirements for equilibrium Molecular Dynamics (MD) simulations in comparison to the time required for sampling with a Boltzmann generator?
>
> The MD simulations for the training data took approximately 1 week to complete. On a single A100 GPU, it takes approximately 20 minutes to sample one million structures. The test set independent simulation took just under 1 week to complete.

---

> > ### Author Response · Authors · 2023-11-19
> >
> > 8. > Have the authors attempted to train the baseline model using their proposed training method?
> >
> > We have added a table in our supplements (Table S.2) that gives results for ablating the training strategies for the baseline NSF architecture. We see that while our training strategy dramatically improves the performance of the NSF architecture, larger systems like villin HP35 and protein G are still not modeled well with the NSF architecture (see Table 1).
> >
> > 9. > What does the energy distribution for HP35 look like? Is it close to the target distribution? What does the reweighted distribution look like?
> >
> > Thank you for your insightful questions. We have added the energy distribution analysis for HP35 in Fig S.2. As with protein G, due to the differences in energy computation and the simulation modality (also discussed in point 1), it is difficult to get an accurate representation of true Boltzmann probabilities for HP35. However, our method is capable of capturing the sampled data distribution very well.

---

> > > ### Comment · Reviewer_3BfG · 2023-11-20
> > > **Clarification for the ESS and reweighting**
> > >
> > > I thank the authors for the detailed additional experiments, I think they improve their work significantly.
> > >
> > > However, it is not quite clear to me how the ESS was computed and how and why they reweight wrt the training energies. Could the authors elaborate on this in more detail? For example, why can't we compute the reweighting weights by computing the energy of the samples (implicit water) and compute the probability of the samples with equation (1)? And then use these to compute the ESS and to reweight. This distribution should then be different from the training distribution. This would moreover help to investigate whether the new discovered state in protein G is indeed stable or not.

---

> > > > ### Author Response · Authors · 2023-11-20
> > > >
> > > > The ESS was computed as in [1]:
> > > >
> > > > For $N$ samples $\mathbf{x}_1, \ldots, \mathbf{x}_N$ from our flow distribution $q(\mathbf{x})$, we assign weights $w_n = \frac{p(\mathbf{x})}{q(\mathbf{x})}, n = 1, \ldots, N$, where $p(\mathbf{x})$ denotes the target distribution. The ESS was then computed as $$ESS = \frac{1}{N \sum\_{i=1}^N \bar{w}_n^2},$$ where $\bar{w}_n$ is the normalizing weight. Typically, we would use the Boltzmann probability $p(\mathbf{x}) \propto e^{-u(\mathbf{x})}$. However, our training distribution itself is a large shift from this distribution, which makes importance sampling with a model trained to maximize the likelihood for the training data difficult (when we apply importance sampling while targeting the Boltzmann probability, only the lowest energy samples receive non-zero weights, which would essentially lead to mode-collapse sampling). This has motivated several recent works that attempt to train normalizing flows in the "data-free" setting [2, 3]. In particular, [3] points out some pitfalls of traditional training of Boltzmann generators when using discrete mini-batches. Unfortunately, [2, 3] suffer from scaling issues as well.
> > > >
> > > > The purpose of our work was to demonstrate plausible methods by which we can scale normalizing flow models for macromolecules. Our model is capable of finding novel conformations for protein G (55 amino acids) and is capable of modeling the training data distribution better than previous models. However, we do acknowledge that there is still much work to be done in exploring the conformational space of larger proteins.
> > > >
> > > >
> > > > [1] Martino et al. Effective Sample Size for Importance Sampling based on discrepancy measures. arXiv:1602.03572, 2016.
> > > >
> > > > [2] Midgley et al. Flow annealed importance sampling bootstrap. ICLR, 2023.
> > > >
> > > > [3] Felardos et al. Designing losses for data-free training of normalizing flows on Boltzmann distributions. arXiv:2301.05475, 2023.

---

> > > > > ### Comment · Reviewer_3BfG · 2023-11-21
> > > > >
> > > > > Thank you for providing additional clarifications, but my question was concerning the target distribution you are reweighting to. You mention that
> > > > >
> > > > > > Typically, we would use the Boltzmann probability $p(x)\propto e^{-u(x)}$. However, our training distribution itself is a large shift from this distribution, which makes importance sampling with a model trained to maximize the likelihood for the training data difficult (when we apply importance sampling while targeting the Boltzmann probability, only the lowest energy samples receive non-zero weights, which would essentially lead to mode-collapse sampling)
> > > > >
> > > > > But it is not clear to me what you do instead.
> > > > >
> > > > > Moreover, it would be interesting what the results are if you use the energy used during KL training as the target energy.

---

> > > > > > ### Author Response · Authors · 2023-11-22
> > > > > >
> > > > > > Thank you for your questions and comments. We apologize for misunderstanding your previous question. In our initial rebuttal analysis, we noticed that if we use true Boltzmann weights, the reweighting would be heavily skewed so as to make only lower energy samples have a non-zero weight. As even our training data would have essentially zero weight, we took a different approach. We assumed that the distribution was still proportional to the exponential of the negative energy distribution, so we used the histogram of $e^{-u(\mathbf{x})}$ for the training data for computing $p_{data}(\mathbf{x})$. We then used the likelihoods of our flow model samples $q(\mathbf{x})$ to generate the importance weights $w(\mathbf{x}) = \frac{p_{data}(\mathbf{x})}{q(\mathbf{x})}$. We further elaborate this in our updated supplements.
> > > > > >
> > > > > > We also updated Sec. F in our supplement to try importance sampling to target the true Boltzmann distribution. We utilize a force field that is closer to our simulating force field and set our target for importance sampling as $p(\mathbf{x}) \propto e^{-u(\mathbf{x})}$. The reweighted distribution is displayed in Fig S.3. As we can see, Boltzmann reweighting tends to sample only for the lowest energy states, and our training data would rarely have nonzero weights. Thus, we took our histogram approach as  mentioned previously.

---

> > > > > > > ### Comment · Reviewer_3BfG · 2023-11-22
> > > > > > >
> > > > > > > Thank you for the explanation, this makes sense to me now.
> > > > > > >
> > > > > > > I concur with the authors regarding the ongoing challenges, particularly in the realm of sampling from the target Boltzmann distribution through reweighting. Nevertheless, I believe their efforts represent a significant stride in the right direction.
> > > > > > >
> > > > > > > In light of this, I have adjusted my rating from 5 to 6.

---

> > > > > > > > ### Author Response · Authors · 2023-11-23
> > > > > > > >
> > > > > > > > Thank you!

---

### Official Review · Reviewer_xhnH · 2023-10-31

**Soundness:** 3 good
**Presentation:** 2 fair
**Contribution:** 3 good
**Rating:** 6
**Confidence:** 3

**Summary:**

The authors create a tractable Boltzmann generator for medium sized protein molecules and show it can recapitulate realistic simulations.

**Strengths:**

Table 1 is very clean and interpretable. Generally, this is an ablation study baked into the main results of the paper, which is helpful.

I think it is great that this method scales to something larger than alanine dipeptide! It makes me sad how frequently this is the only system a method is developed on.

I think Figure 3 is also good to show the examples of proteins that the authors are modeling. I think the figure would be improved by showing the secondary structure on the x-axis of the plot.

**Weaknesses:**

While I think the background section is helpful, sections 2.2 and 2.3 may be helpful to put into context when discussed in the method. While I realize they are described in detail in previous literature, it seems somewhat disjointed when all presented together.

Generally, I need some justification as to why ML models need to be fit to simulation data. Can’t I just run the simulation? What does this give us that a simulation doesn’t?

“For protein G, we use a von Mises base distribution for dihedral coordinates; we noticed that using a von Mises base distribution improved training for the protein G system as compared to a uniform or truncated normal distribution.” Why isn’t this consistent? While I like the Von Mises Distribution, it is odd to me that it isn’t consistent across proteins. As a reader, I want a method I can apply to any protein of interest, or heuristics on which probability distribution to use.

How was the training and test set defined for Protein G? I’d prefer for any notion of novel conformations to be independently confirmed by the simulation. For Figure 4b and 4c, would they never be observed during the simulation?

“An overlay of high-resolution, lowest-energy all-atom structures of protein G generated by the BG model. This demonstrates that our model is capable of sampling low-energy conformations at atomic resolution.” - I don’t think this conclusion is validated by the Figure in 4d. The most you can say is that it kind of looks like the same protein?

**Questions:**

How do “metastable” conformations differ from stable conformations?

For equation 3, for the upper triangle of a distance matrix, over the sum, one can potentially use the (i < j) notation.

“In addition, for larger systems, maximum likelihood training often results in high-energy generated samples.” What does this mean in practice?

Any reason in particular for 58 rotational quadratic spline coupling layers? Or any of the other subsequent number of coupling layers in section 4.1?

Table 1: “Results for ∆D and u(·) that are within tolerable range are bold-faced,” What is “tolerable range”?

Can you define RMSF? I only have a rough understanding of it based of RMSD.

What exactly is unstable during training of these methods? There is a number of works by John Ingraham et al showing instability during training of chaotic systems, particularly of proteins.

Figure 4a - What is the star? I don’t see this defined in the paper.

---

> ### Author Response · Authors · 2023-11-19
>
> 1. > … sections 2.2 and 2.3 may be helpful to put into context when discussed in the method …
>
>     Thank you for the suggestion! If space permits, we will move these sections to be more inline with the discussion of the method in the camera-ready version.
>
> 2. > Can’t I just run the simulation? What does this give us that a simulation doesn’t?
>
>     The primary issue with MD simulations are that they often get stuck in local minima and are unable to sample past large energy barriers. MD simulations are heavily seed-dependent for larger systems, and are computationally expensive. In fact, the total time for training our model and generating one million samples for analysis is less than the time it took to run an independent MD simulation (which we used as our test set in Fig 4). In Sec 4.3, we demonstrate that our model is capable of identifying novel conformations of protein G that are not sampled in all of our simulation data. In particular Fig 4b displays a representative example of a low-energy generated sample not found in the simulation data; this new structure is characterized by a large bent conformation in the hair-pin loop which links beta-strand 3 and 4 of protein G.
>
> 3. > While I like the Von Mises Distribution, it is odd to me that it isn’t consistent across proteins
>
>     We observed an almost negligible difference between the results when using a von MIses distribution, a uniform distribution, or a truncated Gaussian distribution for the base distribution in the ADP and HP35 systems. We hypothesize that this is due to the simplicity of the two systems. In particular, HP35 is characterized primarily by alpha helices, for which backbone dihedrals are relatively fixed. We chose to report the results with the uniform base distribution for these two systems as this is consistent with what has been done in previous works. We hypothesize that for larger, more complex systems, the von Mises base distribution will lead to generally better results. Thus, we suggest that the reader stick with the von Mises base distribution in the general case.
>
> 4. > How was the training and test set defined for Protein G? I’d prefer for any notion of novel conformations to be independently confirmed by the simulation.
>
>     The training for protein G is generated by MD as described in Sec 4.1. An independent simulation was run for the test dataset. We confirmed that the novel conformations as described in Sec 4.3 and Fig 4(b,c) are low-energy structures according to the AMBER force field used for all simulations. While we do not know for sure if these conformations will *never* be observed in simulation, we note that the large bend in the hair-pin loop which links beta-strand 3 and 4 in the new conformation involves crossing a large energy barrier from the closest structure in the simulation dataset.
>
> 5. >  I don’t think this conclusion is validated by the Figure in 4d. The most you can say is that it kind of looks like the same protein?
>
>    Thank you for your observation. We make this statement because the top ten closest generated structures by RMSD to the ground state structure are within 1.5 Angstroms.
>
> 6. > How do “metastable” conformations differ from stable conformations?
>
>     Metastable conformations are functionally important  local energy-minimum structures that are different from global energy-minimum conformation (native structure). Both global and local energy-minimum conformations are stable but their equilibrium probabilities are different. The probability of the various conformations change under different experimental conditions.
>
> 7. > For equation 3, for the upper triangle of a distance matrix, over the sum, one can potentially use the (i < j) notation.
>
>     Thank you for the suggestion! We’ve added this to Definition 3.1.
>
> 8. > “In addition, for larger systems, maximum likelihood training often results in high-energy generated samples.” What does this mean in practice?
>
>     When we train a flow model with internal coordinates with just maximum likelihood training, as is often done in systems as small as alanine dipeptide, the model will often generate outputs that are characterized by clashes (Fig 3d).
>
> 9. > Any reason in particular for 58 rotational quadratic spline coupling layers? Or any of the other subsequent number of coupling layers in section 4.1?
>
>     These are generally arbitrary and chosen based on what fits our GPUs. For our architecture, we utilize 48 GA-RQS backbone coupling layers simply because this is 4 times the number of RQS layers used in [1], and 10 GA-RQS full latent size layers as any more would cause us to run out of memory when training with a batch size of 256. To make the baseline NSF model comparable, we also use 58 RQS layers with a comparable number of parameters for the transformations.

---

> > ### Author Response · Authors · 2023-11-19
> >
> > 10. > Table 1: “Results for ∆D and u(·) that are within tolerable range are bold-faced,” What is “tolerable range”?
> >
> >     Thank you for pointing this out. This is actually a typo. We meant “best results”.
> >
> > 11. > Can you define RMSF? I only have a rough understanding of it based of RMSD.
> >
> >     The RMSF is a measure of the displacement of a particular atom relative to the reference structure, averaged over the number of atoms. Higher RMSF values indicate greater flexibility during the MD simulation. For atom $i$, where $\mathbf{x}_i$ is the coordinates of atom $i$ and $\langle \mathbf{x}_i \rangle$ is the ensemble average position of $i$, the RMSF is given by $$\text{RMSF}_i = \sqrt{\langle (\mathbf{x}_i - \langle \mathbf{x}_i \rangle )^2 \rangle }.$$
> >
> > 12. > What exactly is unstable during training of these methods?
> >
> >     Great question. When we say that training is unstable, we typically refer to gradients exploding or oscillating in such a way that the loss does not converge to a small value.
> >
> > 13. > Figure 4a - What is the star? I don’t see this defined in the paper.
> >
> >     In the caption of Fig 4, for the description of (b), we write “both structures are depicted as stars with their respective structural colors in (a)”. However, there is a typo in (b) regarding the colors. The cyan structure is the novel conformation while the magenta conformation is the closest structure in the training data by RMSD. We have corrected this in the manuscript. We have also updated the zoomed in image that demonstrates the large change in the hair-pin loop between beta strands 3 and 4 to make the conformational difference clearer.
> >
> > [1] Midgley et al. Flow annealed importance sampling bootstrap. ICLR, 2023.

---

### Official Review · Reviewer_2JvV · 2023-10-31

**Soundness:** 3 good
**Presentation:** 3 good
**Contribution:** 2 fair
**Rating:** 5
**Confidence:** 4

**Summary:**

The paper presents a flow architecture and training scheme that improves scalability of Boltzmann generators, allowing application to macromolecules.
The architecture operates on a subset of the internal coordinates (bond angles of side chains are held constant) - which is lower dimensional that the full set of internal coordinates (or cartesian coordinates) while still capturing the most important information on the protein's structure.
The authors introduce a 2-Wasserstein loss which encourages the flow to match the marginal distribution of the distance matrices of the backbone atoms, which leads to more realistic samples.

**Strengths:**

- The authors apply their method to larger molecules than much of previous literature - this is an important step away from toy problems like alanine dipeptide.
- The addition of the Wasserstein loss helps the model generative more realistic molecules that don’t have clashes.

**Weaknesses:**

- Multi-stage training strategy introduces complexity that means significant more effort will be required to tune the algorithm. The third stage of training actually contains two stages. Although the rationale behind the individual losses is clear, the authors do not provide much rationale for each stage of the multi-stage training.
- The architecture is not novel and I do not think this is a significant contribution - internal coordinates have been commonly used, and fixing bond angles of side chains has also been done in literature.

**Questions:**

- What is the rationale/intuition behind each stage of the multistage training? For example, how come the Wasserstein loss is dropped for the final training period? Can this be unified into a single training stage (potentially with annealing of some loss coefficients)?
- The protein-G dataset introduced in the paper seems like it would be useful - will this dataset be released?

---

> ### Author Response · Authors · 2023-11-19
>
> 1. Thank you for your insight and suggestions. Due to the complexity of conformation sampling, we feel that multi-stage training is necessary; the additional complexity is a cost for scaling up to these larger protein systems. What we observed is that NLL training is crucial in order to stably train with the other loss terms. In the literature [1,2], training of Boltzmann generators is done in two stages: 1) minimizing NLL, i.e., maximum likelihood training, and 2) training with a combination of NLL and reverse KL divergence, i.e., training by energy. In Sec 3.5, we highlight several reasons for this:
>
>     - Reverse KL divergence training is expensive and can lead to unstable training without a good initialization.
>     - We introduce the 2-Wasserstein training stage as a way to smooth the transition from pure maximum likelihood training to training by energy.
>     - The 2-Wasserstein loss is dropped at the end; this final stage can be seen as an energy “relaxation” stage.
>
>     We provide a summary of the hyperparameters in the supplements (Table S.1). Indeed, we could try to unify everything into a single training stage with a loss coefficient schedule, e.g., annealing. However, our previous attempts to move in this direction were not as successful as training in this multi-stage manner.
>
> 2. Indeed, internal coordinates have been used in the past, as we have mentioned in Sec 3.2. However, they have not been utilized for Boltzmann generators in any system beyond alanine dipeptide. The key contributions of our architecture are: 1) GAUs, 2) rotary positional embeddings and the T5 relative positional bias for learning long range interactions, 3) split backbone and sidechain channels for efficient training. These three contributions combined allowed our model to successfully learn long range interactions for internal coordinates. We have added a table in our supplements (Table S.2) that gives results for ablating the training strategies for the baseline NSF architecture. We see that while our training strategy dramatically improves the performance of the NSF architecture, larger systems like villin HP35 and protein G are still not modeled well with the NSF architecture (see Table 1).
>
> 3. We will indeed release the protein G dataset upon acceptance. Thank you for the suggestion.
>
>
> [1] Noe et al. Boltzmann generators: sampling equilibrium states of many-body systems with deep learning. Science, 2019.
>
> [2] Midgley et al. Flow annealed importance sampling bootstrap. ICLR, 2023.

---

> > ### Comment · Reviewer_2JvV · 2023-11-20
> >
> > Thank you for your reply. It is great to hear that you will be releasing the dataset!
> > Note that the reference [2] that you cited in your comment does not follow the two stage training as you stated, and uses a single stage of energy-based training.

---

> > > ### Author Response · Authors · 2023-11-20
> > >
> > > Thank you for your comments. We apologize for the incorrect citation.

---

### Official Review · Reviewer_VZgX · 2023-10-31

**Soundness:** 4 excellent
**Presentation:** 2 fair
**Contribution:** 3 good
**Rating:** 6
**Confidence:** 3

**Summary:**

Based on previous works which use normalizing flows to generate distribution of conformations of proteins, this paper extends the scalability of NFs by (1) using internal coordinates instead of Cartesian coordinates (2) introducing 2-Wasserstein distance as training loss. (3) introducing new architecture. The proposed methods were proved useful in a small "protein" ADP and two larger proteins, in terms of distance distortion, energy and NLL loss.

**Strengths:**

This paper proposed to use internal coordinates instead of Cartesian coordinates, and split the backbone and side chain. From my point of view, this is the correct way to do protein conformation generation, since the Cartesian coordinates is redundant and highly correlated, whereas the internal coordinate is more compact and less correlated. I like this idea.

**Weaknesses:**

Please refer to the "questions" section.

**Questions:**

1. I didn't fully understand the exact difference between the proposed "new architecture" and NSF. Is GAU the difference, or something else? Could you illustrate the difference more explicitly. Thanks.
2. The experiment results are very good, but it is better to understand what is the key factor for the improvement by more ablation studies, such as NSF+NLL+KL+W2, Cartesian coordinate+your architecture+NLL+KL+W2, etc.
3. In Table 1, I notice that in Protein G and HP35, some energy are as large as 10^6 to 10^10. What are the reasons for these extremely high energies? If they are caused by some naive reasons, can we quick-fix them to make the comparison more fair?
4. In Table 1, NLL loss of Protein G is negative, why?

---

> ### Author Response · Authors · 2023-11-19
>
> 1. The new architecture refers to several changes: 1) GAUs, 2) rotary positional embeddings and the T5 relative positional bias for learning long range interactions, 3) split backbone and sidechain channels for efficient training. These three contributions combined allowed our model to successfully learn long range interactions for internal coordinates. In addition, we make use of reduced internal coordinates for representing our systems, which are notoriously more difficult to handle in the BG literature. We demonstrate that our model (architecture + training strategies) is capable of handling the difficulties of working with internal coordinates. The NSF baseline model which we used for comparison is the same as used in previous literature [1,2].
> 2. We thank you for recognizing our experiment results. We noticed that our model architecture achieved much better results with NLL training than the baseline NSF model. We have added a full set of ablation results with the NSF architecture in the supplements (Appendix Sec. E, Table S.1). As we can see, while our different training strategies improve upon the baseline model with NLL training, the improvements are not as drastic as for our architecture. As for Cartesian coordinates, we purposefully utilize a reduced set of internal coordinates in order to reduce the dimension of the problem (Sec 3.2).
> 3. These large energies are typically caused by van der waal’s interactions, i.e., atomic clashes (Sec 4.2). As we are modeling our inputs with internal coordinates, it is expected for training to result in clashes more than if we were to model with Cartesian coordinates (Sec 3.2). However, a key contribution in this work is the introduction of a training strategy and architecture that is capable of handling such issues that are typical when working with internal coordinates. Before submitting this work, we tried several methods to reduce the clashing in a fair manner such as by incorporating a violation loss as in AlphaFold2 [3]. However, these methods proved ineffective compared to the contributions in this paper for our internal coordinate inputs.
> 4. For protein G, we use a von Mises base distribution for rotatable coordinates as this led to better empirical results (Sec 4.1). The continuous probability densities for computing the log likelihoods in the base distribution result in positive log likelihood values (negative NLL). In comparison, we use a uniform base distribution for ADP and HP35.
>
>
> [1] Kohler et al. Flow-matching - efficient coarse-graining molecular dynamics without forces. arXiv:2203.11167, 2022.
> [2] Midgley et al. Flow annealed importance sampling bootstrap. ICLR, 2023.
> [3] Jumper et al. Highly accurate protein structure prediction with AlphaFold. Nature, 2021.

---

### Meta-Review · Area_Chair_XWVX · 2023-12-06

**Metareview:**

This paper introduces a novel flow architecture employing split channels and gated attention to efficiently learn the conformational distribution of proteins defined by internal coordinates. Experiments on macromolecules, such as proteins, are conducted to evaluate the performance of the model. While the paper is interesting, well-organized, and easy to follow, the novelty of the methodology is considered not significantly impactful. The rebuttal has addressed most concerns raised by reviewers, resulting in an overall borderline rating. However, during the discussion, no reviewer strongly supports the paper. The Area Chair and reviewers acknowledge the valuable contribution in scaling up the Boltzmann generator to large proteins but express the view that the novelty of the paper is marginal. Considering these factors, the Area Chair recommends rejecting the paper at the current stage.

**Justification For Why Not Higher Score:**

The contribution of the method is not very significant, leading to a recommendation for a borderline reject.

**Justification For Why Not Lower Score:**

NA

---

### Decision · Program_Chairs · 2024-01-16

Reject